# Life Cycle Impact Assessment of Load-Bearing Straw Bale Residential Building

**DOI:** 10.3390/ma14113064

**Published:** 2021-06-04

**Authors:** Rozalia Vanova, Michal Vlcko, Jozef Stefko

**Affiliations:** Department of Wood Structures, Faculty of Wood Sciences and Technology, Technical University in Zvolen, T. G. Masaryka 24, 960 01 Zvolen, Slovakia; michalvlcko@hotmail.com (M.V.); stefko@is.tuzvo.sk (J.S.)

**Keywords:** life cycle impact assessment, damage assessment, single score, construction materials, straw bale

## Abstract

As a renewable raw material, straw bale represents a sustainable way of construction with minimal environmental impact. This paper focused on life cycle impact assessment of load-bearing straw bale residential building. Product stage from raw materials extraction to manufacture of construction materials was considered in the assessment including seven variations of straw bale. Construction materials were evaluated due to IMPACT 2002+ method. Both midpoint and endpoint impact categories were included. The results showed the importance of straw bale origin. Ecosystem quality impact of straw from extensively cultivated pastures was twenty times higher than that of intensive crop production, thus making a significant difference to an overall score of the construction. Results showed advantage of straw as a construction material particularly when used locally. In addition, significant contributions of other construction materials were identified.

## 1. Introduction

### 1.1. Environmental Impact and Life Cycle Assessment

From 1990 to 2014 industrialization caused an increase of direct emissions, including energy and non-energy process emissions by 65% [1], thus forming a difficult situation we are facing today. The World Economic Forum [2] claim climate-related natural disasters such as hurricanes, droughts, and wildfires are becoming more intense and more frequent. Together with faster polar ice melting threatening coastal populations, we are forced to take action to avert irreversible changes of our environment. Andrew Boyd, a co-founder of Climate Watch project, stated we have just over 7 years to help our planet restore [3].

A way how to survive these changes is to switch on sustainable raw material and energy sources as well as to create technologies preventing waste production as much as possible. Consequently, in 2006 the International Organization for Standardization released a framework for Life Cycle Assessment—LCA methodology [4,5], providing principles and guidelines of products and services impact assessment over the whole life cycle. This method has been being commonly used among researchers worldwide and brings progressive view on environmental pollution caused by current technology and society requirements.

LCA method helps us understand products and services within their individual life cycle stages. According to EN 15804 [6] buildings and construction materials have four main stages—product, construction, use and end-of-life stage; and one additional stage expresses possible benefits and loads beyond system boundaries (Figure 1). Each stage may have a different impact. In order to choose the product or service with the lowest environmental impact, it is crucial to identify and understand its impacts throughout the whole life cycle.

The LCA evaluates input and output material and energy flows within the system under study by midpoint and/or endpoint impact categories depending on a specific calculation method. Figure 2 illustrates relation between midpoint and endpoint categories on the basis of IMPACT 2002+ calculation method [7]. Characterization stage as the key step of Life Cycle Impact Assessment (LCIA) can be supplemented by damage assessment, weighting, or single score analysis. These steps serve to compare the magnitude of selected impact categories or products. The results provide the decision makers with information on environmental benefits and burdens connected with a product or service.

### 1.2. Straw as a Construction Material

Straw has been used as a construction material for thousands of years. Its chemical composition is comparable to wood biomass and can be implemented into construction in various ways. Common types of straw-based buildings present load-bearing or post and beam construction. Nowadays, straw-based biocomposites are the scope of interest [8].

Straw-based constructions represent a green variant of conventional construction materials. Muazu et al. [9] affirmed traditional constructions utilized local raw material sources and provided simple processing and minimal transport requirements. Bhattarai et al. [10] in their study highlighted environmental issues of straw bales. Ashour and Wu declared [11] it is possible to use straw bales for construction almost any structure including thermal and acoustic insulation as well as fire safety walls. Properly designed straw-based constructions can meet thermo-technical criteria and environmental requirements (Table 1).

Potential reduction in waste disposal and general availability of straw bales make straw a remarkable construction material. The exact number of straw-based constructions in the world is unknown. However, there are databases that register these buildings, e.g., the European Straw Building Association (ESBA) in Europe [21]. According to Cornaro et al. [22] straw-based buildings were mostly spread in the USA, Western Europe, and Asia.

Focusing on straw bales throughout their life cycle, Gonzalez [23] reported that the embodied energy of straw bales was lower by 5.8% than that of fired brick and 19.8% lower in case of cement block. Moreover, he proved that transport distances of common masonry construction materials considerably increased overall environmental damage, highlighting a relevant role of renewable agricultural products in lowering impacts of construction materials, as well as options for sustainable production.

A study by Abd-Elhafeez et al. [24] comparing straw bale and masonry brick construction showed that straw bale construction reached 40% lower discomfort degree hours, 82% lower energy consumption, and 80% lower CO_2_ emissions than masonry brick construction.

Findings of Cornaro et al. [22] revealed production and construction of straw-based wall accounted for 50% lesser embodied energy and CO_2_ emissions than the masonry wall. Moreover, the use stage of straw-based wall was responsible for about 91% of the total embodied energy and 93% of the total CO_2_ emissions compared to masonry wall liable to 85% and 88% of the total embodied energy and CO_2_ emissions, respectively.

Nozdrovicky et al. [25] investigated technological effects of straw baling technology and found out smaller field acreage required frequent machine turning which increased the working time of the machinery which led to higher CO_2_ emissions.

International Energy Agency [26] reported buildings and construction industry was responsible for 36% of the total final energy consumption and 38% of the total energy-related CO_2_ emissions in 2019.

International Energy Agency. Global Status Report for Buildings and Construction 2019. IEA, Paris. Available online: https://www.iea.org/reports/global-status-report-for-buildings-and-construction-2019 (accessed on 16 January 2021). This paper identifies product stage environmental impacts of straw bale building considering several types of straw bale. Midpoint (carcinogens; non-carcinogens; respiratory organics and inorganics; ionizing radiation; ozone layer depletion; aquatic and terrestrial ecotoxicity; terrestrial acidification and nitrification; land occupation; aquatic acidification and eutrophication; global warming; non-renewable energy; mineral extraction) and endpoint impact categories (human health, ecosystem quality, climate change, resources) are introduced to thoroughly explore impacts of used construction materials.

## 2. Materials and Methods

The selected load-bearing straw bale construction was assessed in terms of its environmental impact, based on the LCA methodology [4,5] and IMPACT 2002+ calculation method [7]. Stages A1 to A3 were included in the assessment (Figure 1) [6]. Functional unit was characterized by the whole straw bale construction representing 76.05 m^2^ of gross internal area. The construction was proposed by authors denoting typical way of construction object for small family housing (Figure 3).

The data on components were taken from building area statement. Thermal characteristics (specific heat demand and heat transfer coefficient) were calculated due to Slovak technical standard STN 73 0540-3 [27]. Specific heat demand for heating was set to 31 kWh/m^2^a. Primary energy demand was not taken into account.

Material composition (Table 2) and weight distribution within the construction (Table 3) were described in their respective tables. Environmental impact was expressed via midpoint and endpoint impact categories.

Market for standalone straw production database including transport was chosen for the initial assessment of the construction. In addition, six more straw databases were assessed separately (Table 4) in order to identify the most environmentally beneficial variant of straw. Selected straw databases were defined as co-products of wheat production and differed in geographical locations, type of cultivation along with area of occupation and transformation of soil [28]. Construction material no. 10 (Table 3) referred to Straw G database (Table 4).

## 3. Results

At first, the proposed construction was assessed by the characterization stage. All material and energy flows were assigned to respective midpoint impact categories. The characterization result (Table A1) showed foam glass hit environment the worst at seven impact categories—carcinogens (74.23 kg C_2_H_3_Cl eq), respiratory inorganics (8.42 kg PM2.5 eq), ionizing radiation (59.14 × 10^3^ Bq C-14 eq), terrestrial acidification/nitrification (98.11 kg SO_2_ eq), aquatic acidification (25.78 kg SO_2_ eq), global warming (58.93 × 10^2^ kg CO_2_ eq), and non-renewable energy (90.45 × 10^3^ kg oil eq).

OSB affected the worst respiratory organics (1.72 kg C_2_H_4_ eq). Metal sheet roofing was indicated as the worst material in mineral extraction responsible for 555.82 kg Fe eq resource depletion, representing only 0.25% of construction weight.

Despite relatively low weight contribution (0.9%), XPS achieved the highest negative impact in ozone layer depletion category (1.15 × 10^−3^ kg CFC-11 eq) due to the extrusion process where HFC-134a and HFC-152a were still being used as blown agents besides CO_2_.

Standalone straw achieved the highest negative impact in land occupation (61.34 × 10^3^ m^2^ eq organic arable land in a year) and aquatic eutrophication (3.08 kg PO_4_^3−^ eq) impact categories. Nonetheless, it reached negative values in non-carcinogens (−14.29 × 10^2^ kg C_2_H_3_Cl eq), aquatic ecotoxicity (−27.30 × 10^5^ kg TEG eq), and terrestrial ecotoxicity (−35.61 × 10^5^ kg TEG eq) impact categories proving positive effects on environment related to the incorporation of pollutants into its structure. However, the negative impact could occur at the end of the life cycle when pollutants could revert to the environment. In the category of land occupation, the impact was caused by the process of standalone straw production, especially by inputs from nature “Occupation, pasture, artificial, intensive”. The same process was responsible for negative impact on aquatic eutrophication caused by phosphates leaching. On the other hand, zinc and copper accounted for positive effect since their deposition into the straw structure.

After characterization results, damage assessment was carried out (Figure 4). Damage assessment constituted four endpoint impact categories (human health, ecosystem quality, climate change, and resources) formed by multiplying the midpoint characterization potentials with the damage characterization factors of the reference substances based on Jolliet et al. [7]. Human health was introduced as disability-adjusted life years (DALY). The damage to biodiversity reflected the fraction of species that has been lost in comparison with a natural or undisturbed area as the potentially disappeared fraction of species over a certain area during a certain time. Resources were characterized by the amount of additional primary energy required per unit of mineral and of total non-renewable primary energy for energy carriers. Climate change remained the same unit as for midpoint impact category.

Positive effect of straw (no. 10) heavy metals absorption from soil resulted in human health negative values from the damage assessment point of view (Figure 4). Foam glass (no. 12) confirmed the worst values in climate change, resources and human health impact category. Ecosystem quality was mostly affected by straw bale due to the intensive pasture occupation.

Concrete (no. 2), fiberboard (no. 3), and clay plaster (no. 4) showed similar values in almost all categories except climate change dominated by concrete (Figure 4). Mastic asphalt (no. 5) and roof insulation foil (no. 8) had a negligible effect on the environment due to their low weight contribution (0.05% and 0.01%, respectively) within the construction.

Eventually, single score endpoint impacts were identified (Figure 5). Single score assessment evaluates contribution of each endpoint impact category on the whole material. One point (Pt), as the single score unit, stands for one-thousandth of the yearly environmental load of one average European inhabitant.

Single score results clearly identified straw bale (no. 10) as the biggest burden for the environment mainly because of intensive pasture occupation as already mentioned above. Moreover, it should be stated straw represented only 15.09% of weight within the whole construction. The second worst material remained foam glass (no. 12) followed by metal sheet roofing (no. 13). Moreover, metal sheet roofing had more than 16 times lower weight compared to foam glass and represented nearly 55% of the foam glass overall endpoint impact.

In order to investigate the contribution of straw origin to overall environmental impact six more straw databases were assessed (Table 4 and Table 5; Figure 6). Straw G, identical to construction material no. 10 used in the initial assessment, affirmed to be the most harmful straw alternative despite positive effects on human health. On the other hand, Straw C corresponding to intensive wheat production appeared to be the most environmentally beneficial option the least affecting ecosystem quality (4.98% out of overall Straw G impact in respective category).

According to the single score assessment of different straw alternatives (Figure 6), replacing Straw G (2.65 Pt) with Straw C (0.29 Pt) would lead to decrease in potential environmental impact of construction straw bale by 89.06%. Though, Straw C does not operate with market transportation distances. Nevertheless, if locally available sources of straw bale were used, the transportation impact would be negligible.

## 4. Discussion

Databases used for the assessment were part of global datasets referring to average technology and transport distances from all over the world. The study proved importance of straw origin on the final impact of the construction and supported the findings of Haas et al. [29] who claimed extensive and organic farming reduced negative environmental impact compared to intensive grassland farming. Each straw alternative considered in our study had specific relative percentage endpoint category impact (Table 5) closely related to specific technological operations according to the cultivation type. That proved the relevance of database selection when performing LCA and importance of locally available sources as well as contribution of individual cultivation type. As the best option for construction straw bale, Straw C was chosen representing intensive wheat production in Swiss. Foam glass, metal sheet roofing, wood flooring, and XPS were identified to bear the majority of the building impact on environment. Waterproofing insulation environmental impact was denoted as negligible.

Global warming midpoint impact category of the SBRB was 304.93 kg CO_2_ eq/m^2^ and 294.35 kg CO_2_ eq/m^2^ in the case of Straw C alternative. Wood-based single-family building assessed by Petrovic et al. [30] produced 169.80 kg CO_2_ eq/m^2^ throughout the product stage. Other assessment of wood-based building done by Mitterpach et al. [31] showed impact of 389.62 kg CO_2_ eq/m^2^. However, each construction was unique in terms of material composition, weight distribution, number of floors, floor area, inventory used for the assessment, calculation method, geographical region and many other factors. Most of the studies contained data specific for the research, making it non-comparable with other records. Such constraints therefore brought different values and did not reflect reality. Moreover, quantity of construction materials within construction was often estimated, resulting in further inaccuracies.

The findings of Moschetti et al. [32] pointed towards the need for a stronger strategic emphasis on the embodied energy and emissions of materials for a successful transition from zero-energy to zero-emission building target. Despite relatively high impact on ecosystem quality straw remained an environmentally beneficial construction material.

As well as Takano et al. [33] compared different databases, our results of the specific impacts on the environment were always individual and different according to the chosen calculation method and also within individual impact categories. Furthermore, attributive and consequential approaches could have different results, as was proved by Finnveden et al. [34]. Moreover, further research on environmental characteristics of straw constructions could be made considering straw as an agricultural waste material and its effect on the total score. The study did not take into account economic factor, environmental conditions and durability of the structure.

## 5. Conclusions

This paper revealed the environmental impact of straw bale residential building (SBRB) at midpoint and endpoint impact categories and proved the importance of suitable construction materials selection. The study also identified environmentally inappropriate and burdensome materials.

To express possible impacts of different straw bales origin on construction embodied environmental impact seven variations of straw were considered. Results pointed at vast differences between individual products, especially depending on the production technology, tillage, geographical range, and transport distances. Proper selection of straw bale could cut the total embodied emissions up to 89%. In all cases, straw had the greatest impact on ecosystem quality due to land occupation. Intensive wheat straw production required more fertilizers, thus supported eutrophication. On the other hand, vast land occupation done by extensive production of standalone straw considerably damaged the biodiversity, but remained positive in non-carcinogens, aquatic and terrestrial ecotoxicity impact categories due to absorption of heavy metals during biomass growth.

Understanding the environmental impact of construction materials presents a key point of construction design. However, to build the most environmentally beneficial construction it is necessary to look at each life cycle stage in detail and assess them as a whole. In these terms, further analysis should be made to evaluate the overall life cycle environmental impact of such structure considering several variations of the most burdensome construction materials and energy sources.

## Figures and Tables

**Figure 1 materials-14-03064-f001:**
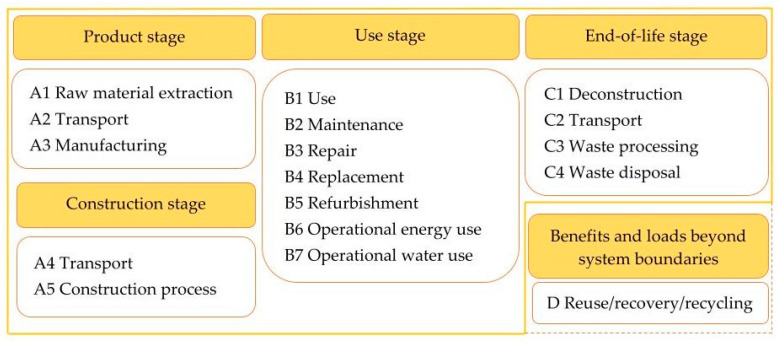
Life cycle stages of buildings.

**Figure 2 materials-14-03064-f002:**
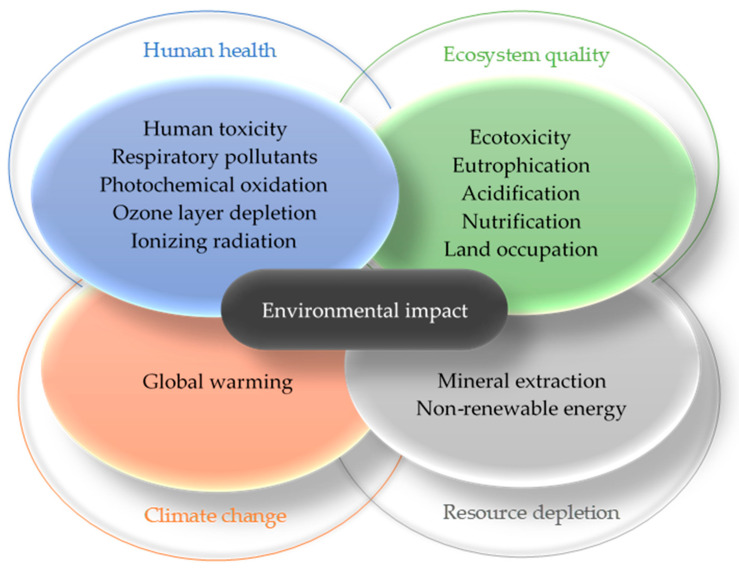
Example of possible environmental impacts and their relation to midpoint and endpoint impact categories.

**Figure 3 materials-14-03064-f003:**
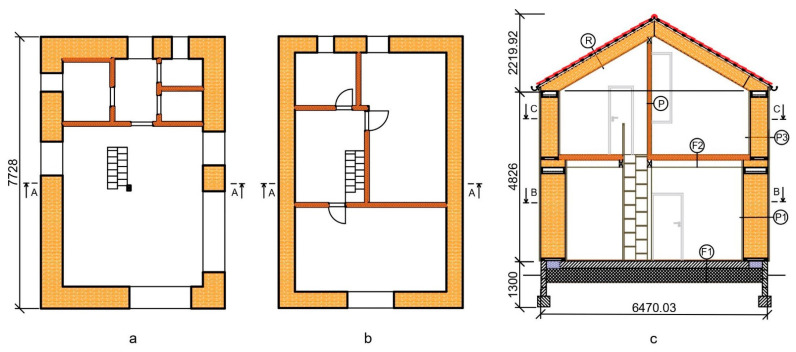
Cross sections of the straw bale residential building (SBRB): (**a**) First floor section B-B; (**b**) Second floor section C-C; (**c**) Cross section A-A. Material composition of marked structures were given in Table 2. Measures are given in milimetres.

**Figure 4 materials-14-03064-f004:**
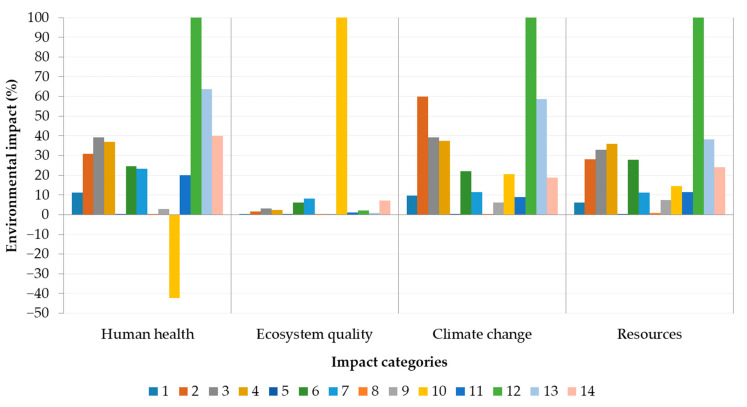
Damage assessment of construction materials within each endpoint impact category (human health, ecosystem quality, climate change and resources). Numbers refer to construction materials listed in Table 3.

**Figure 5 materials-14-03064-f005:**
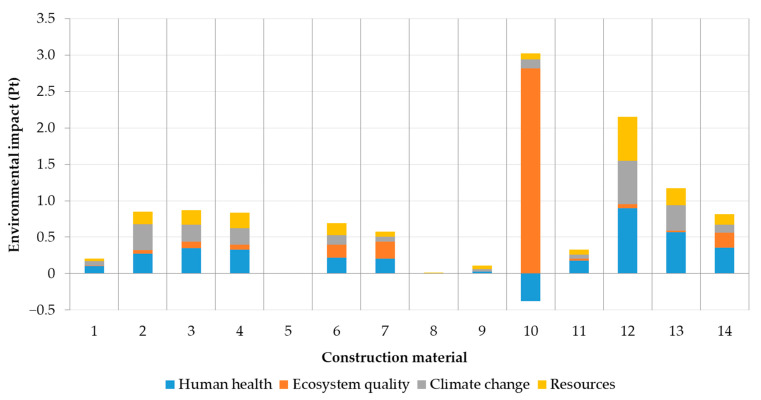
Contribution of each endpoint impact category (human health, ecosystem quality, climate change and resources) to the environmental impact of construction materials (Table 3)—a single score assessment. Numbers refer to construction materials listed in Table 3.

**Figure 6 materials-14-03064-f006:**
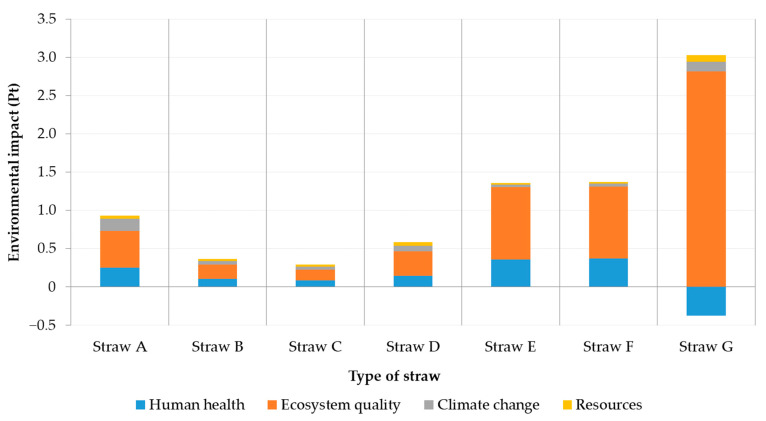
Contribution of different straw alternatives to environmental impact—a single score assessment.

**Table 1 materials-14-03064-t001:** Selected characteristics of straw.

Characteristics	Argument
Fire Resistance	Compressing the straw into a dense block dramatically decreases the ability of oxygen to catch fire. A study by Dzidic [12] showed load-bearing wall made of straw bales, plastered on both sides, loaded with 12 kN/m resisted to standard fire for 2 h 26 min, reaching the REI 120 fire resistance. For comparison, wall panel framed with wooden studs and insulated by straw bale, loaded with 20 kN/m achieved the REI 60 fire resistance.
Thermal insulation	Advantages of straw as a thermal insulation material were proved by Petkova-Slipets and Zlateva [13] embracing TCi Mathis Thermal Conductivity Analyser for non-destructive measurement of thermal-physical characteristics. They concluded even less than 0.5 weight % of straw in a composite material contribute to more than 50% increase in specific heat capacity and up to 56% decrease of the thermal conductivity. Costes et al. [14] measured the thermal conductivity of straw bales in Guarded Hot Plate apparatus. They compared performance changes of 36 to 46 cm wall width and 20 to 70 kg/m^2^ straw mass. Thermal conductivity of considered straw bale walls ranged between 0.056 and 0.097 W/mK. Sabapathy and Gedupudi [15] highlighted orientation of straw fibers could influence the thermal transport properties. The thermal conductivity of parallel oriented straw samples (0.069–0.194 W/mK) was higher than random and perpendicular oriented straw samples (0.040–0.084 W/mK).
Acoustic insulation	Measurements of Teslík et al. [16] on airborne insulation properties of different types of straw-based walls turned out surface treatments had the largest share in the value of airborne soundproofing. Moreover, each investigated wall met the requirements of the Weighted Sound Reduction Index for partition structures in EU (Rw = 48–57 dB). Cascone et al. [17] in their study compared acoustic performances of timber-framed walls with straw bale insulation to similar walls containing expanded polystyrene. The results showed straw bales as a better sound insulation material referring to the Weighted Apparent Sound Reduction Index (R‘w) value of 49 dB and the Weighted Standardized Level Difference of the façade (D_2m,nt,w_) equaled to 43 dB.
Durability, moisture and biological resistance	Regarding moisture as one of the factors of construction durability, Koh and Kraniotis [18] indicated exterior climate as the most important factor of hygrothermal characteristics of straw bale constructions. Exterior side of the straw-based walls were the most susceptible to mold growth. Nonetheless, they suggested application of exterior cladding with ventilated air gap could prevent moisture penetration. The risk of fungal and mold development in straw was confirmed by Marques et al. [19]. At the same time, they claimed proper design and construction of a straw bale building is essential to ensure durability and prevent biological development.
Toxicity	Allam et al. [20] reminded burning of straw releases large amounts of air pollutants that could cause serious environmental problems. Straw bales remain intact in the form of construction material. Therefore, upcycling straw into a construction material could prevent emission release from burning.

**Table 2 materials-14-03064-t002:** Material composition and heat transfer coefficient of the SBRB.

Structure	Construction Materials	Heat Transfer Coefficient [W/(m^2^K)]
Flooring 1st floor F1	Wood flooring 22 mm Fiberboard insulation 8 mm Reinforced concrete 200 mm Damp proofing 8 mm Foam glass 400 mm Geotextile 1.5 mm	0.174
Peripheral wall 1 P1	Clay plaster 40 mm Wheat straw bale 700 mm Clay plaster 40 mm	0.073
Peripheral wall 2	Clay plaster 40 mm Wheat straw bale with stalks parallel to the direction of heat flow 500 mm Clay plaster 40 mm	0.153
Peripheral wall 3 P3	Clay plaster 40 mm Diagonal lathing 20 mm × 100 mm Double columns 60 mm × 140 mm Wheat straw bale 360 mm Clay plaster 40 mm	0.153
Flooring 2nd floor F2	Wood flooring 24 mm Timber grate 50 mm × 100 mm Fiberboard insulation 100 mm Fiberboard insulation 8 mm Timber plate 24 mm Joist 140 mm × 240 mm	-
Partition P	Clay plaster on reed grate 30 mm Oriented strand board 3–15 mm Column 60 × 40 mm Fiberboard insulation 8 mm OSB 3–15 mm Clay plaster on reed grate 30 mm	-
Roof R	Timber plate 20 mm Rafter 360 mm × 60 mm Wheat straw bale 360 mm Diffusion foil Lathing 50 mm × 40 mm Metal sheet roofing	0.169

**Table 3 materials-14-03064-t003:** Weight distribution of the construction materials based on the actual area statement of the construction. Each material was given a number for further assessment. Database name referred to a specific dataset used in the evaluation. Allocation at the point of substitution (APOS) and unit processes (U) were selected for the life cycle modelling.

No.	Material	Ecoinvent Database	Weight (kg)	Weight (%)
1	Reinforcing steel	Reinforcing steel (GLO)|market for|APOS, U	241.49	0.31
2	Concrete	Concrete, 20 MPa (GLO)|market for|APOS, U	33,849.56	43.20
3	Light density fiberboard (LDF)	Fiberboard, soft, without adhesives (GLO)|market for|APOS, U	2068.39	2.64
4	Clay plaster	Clay plaster (GLO)|market for|APOS, U	20,127.30	25.69
5	Mastic asphalt	Mastic asphalt (GLO)|market for|APOS, U	42.84	0.05
6	Oriented strand board (OSB)	Oriented strand board (GLO)|market for|APOS, U	2492.35	3.18
7	Timber	Sawnwood, beam, softwood, raw, dried (u = 10%) (GLO)| market for|APOS, U	2692.13	3.44
8	Roof insulation foil	Roof insulation foil (Own suggestion)	9.92	0.01
9	Extruded polystyrene (XPS)	Polystyrene, extruded (GLO)|market for|APOS, U	66.85	0.09
10	Straw bale = Straw G	Straw, stand-alone production (GLO)| market for|APOS, U	11,823.12	15.09
11	Particleboard	Particle board, for indoor use (GLO)| market for|APOS, U	573.30	0.73
12	Foam glass	Foam glass (GLO)|market for|APOS, U	3266.15	4.17
13	Metal sheet roofing	Aluminum, wrought alloy (GLO)|market for|APOS, U	192.35	0.25
14	Wood flooring	Wood flooring (Own suggestion)	915.50	1.17

**Table 4 materials-14-03064-t004:** Straw databases used for straw environmental contribution comparison. Land occupation described the duration of land use from soil cultivation until harvest and the yield per area unit in a year. Land transformation referred to the area required to produce 1 kg of straw.

Type of Straw	Ecoinvent Database	Type of Land Use	Land Occupation Area (m^2^)	Land Transformation Area (m^2^)
Straw A	Straw (AU)|wheat production|APOS, U	Annual crop	0.4719	0.4706
Straw B	Straw (CH)| wheat production, Swiss integrated production, extensive|APOS, U	Annual crop, non-irrigated, extensive	0.1299	0.1636
Straw C	Straw (CH)|wheat production, Swiss integrated production, intensive|APOS, U	Annual crop, non-irrigated, intensive	0.1073	0.1350
Straw D	Straw (RoW)| wheat production|APOS, U	Annual crop	0.1880	0.1880
Straw E	Straw, organic (CH)|wheat production, organic|APOS, U	Annual crop, non-irrigated, intensive	0.0737	0.0734
Straw F	Straw, organic (RoW)|wheat production, organic|APOS, U	Annual crop, non-irrigated, extensive	0.1607	0.2023
Straw G	Straw, stand-alone production (GLO)|market for|APOS, U	Annual crop, non-irrigated, extensive	0.1607	0.2023

**Table 5 materials-14-03064-t005:** Relative percentage comparison of individual straw alternatives environmental impact. For each impact category the worst and the best straw alternative was highlighted according to the highest contribution within individual category.

Straw Database	Environmental Impact (%)			
	Human Health	Ecosystem Quality	Climate Change	Resources
Straw A	66.94	16.95	100.00	46.92
Straw B	26.98	6.75	26.95	35.74
Straw C	21.80	4.98	25.04	33.42
Straw D	37.78	11.46	42.31	54.23
Straw E	95.35	33.43	20.79	26.44
Straw F	97.60	33.43	23.17	26.54
Straw G	−100.00	100.00	75.24	100.00

## Data Availability

Additional data can be found at https://www.ecoinvent.org/database/older-versions/ecoinvent-34/ecoinvent-34.html (accessed on 2 June 2021).

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
