# Peer review of "Life Cycle Impact Assessment of Load-Bearing Straw Bale Residential Building"

_materials, 2021, doi:10.3390/ma14113064_

Round 1

Reviewer 1 Report

Dear authors,

Thank you for the submitted comprehensive study of the environmental impact assessment of a straw bale of an apartment building with regard to other materials used.
The work is written matter-of-factly and clearly.
This may be an advantage, but a longer introduction showing the current state of construction of straw bales in the world would be appropriate.
The principle of LCA is relatively well known and therefore I consider the explanation to be sufficient.

I appreciate the extensive discussion and comparison with other articles.
The conclusions are fine.

A few notes on the format:
Caution: you must use the dot as a decimal separator in numbers.
If you insist on using graphs in Excel (it's very unscientific) you need to change the font to Palatino Linotype. Your article is now inappropriate.
The tables are diverse and inconsistent, sometimes it is necessary to fine-tune their size.
There are language ambiguities and it would be appropriate to use language editing.

Regards,

Author Response

Dear rewiever,

at first, I deeply apologise for time delay. Thank you for your report. A major revision was done taking into account all review reports. We edited all points according to your report. Moreover, references; technical details and a picture was added.

Yours sincerely,

Rozália Vaňová

Reviewer 2 Report

The submitted article “materials-1176878_v1” entitled: “Environmental Impact Assessment of Load Bearing Straw Bale Residential Building” is an effort to investigate the life cycle impact assessment of load bearing straw bale residential building. The presented study slightly falls within the scope of Materials Journal. Further, the submitted manuscript has serious flaws and the overall quality, including value of contribution, methodology, analysis and presentation style is low for a typical journal paper. The following issues are some sample weaknesses:

The literature background provided is shallow, lacking coherence and failing to establish the relevance of the work reported in the paper. The state-of the-art does not adequately highlight the gaps in order to promote the research significance and the objectives of the paper. Research significance, novelty and subsequent impact of the study on the state of the practice is not adequately highlighted. Presentation is rather brief and important technical details are missing. Methodology, description of the properties of the materials and the analysis is rather basic. In general, contents of the paper seem poor to be worthy of publication. Manuscript is not-well written. It lacks of a deep scientific description of the concept, while the results are not discussed in-depth. The authors should make a significant effort to re-write the article in a way that would really be useful for the scientific community. Paper does not really bring to bear any significant new scholarly contributions and does not appear to offer any noticeable improvement.

Author Response

Dear rewiever,

at first, I deeply apologise for the time delay. Thank you for your report. A major revision was done taking into account all review reports. References, technical details and a picture was added. Nearly all parts of the study were edited. We believe we met all of your requirements. 

Yours sincerely,

Rozália Vaňová

Reviewer 3 Report

Dear authors, i attach a pdf with some comments. Please in case of resubmission provide your answers on the same file to speed up a second review.

Author Response

Dear rewiever,

at first, I deeply apologise for the time delay. Thank you for your report. A major revision was done taking into account all review reports. Please, see the attachment.

Yours sincerely,

Rozália Vaňová

Reviewer 4 Report

The article submitted for review is of a popular scientific nature. It concerns a niche issue of implementing residential buildings with the use of straw elements. There is no unequivocal leitmotif in the work: many issues are discussed superficially, but without detailed analysis. In practice, it is not known which building is the subject of the analysis: there are no drawings in the form of projections and sections. The work does not specify solutions for structural elements. The formulated conclusions do not take into account the economic factor, environmental conditions and issues related to the issues of durability of building structures. 

Recommendations:                                                                                            - one technical issue should be focused, not a social one, and analyzed in detail,                                                                                                                 - it should be clearly indicated whether this type of research has any future or whether it is an eminently theoretical direction of research 

Author Response

Dear rewiever,

at first, I deeply apologise for the time delay. Thank you for your report. A major revision was done taking into account all review reports. References, technical details and a picture was added. Nearly all parts of the study were edited. 

The study did not take into account economic factor and environmental conditions, because these factors can vary depending on a country. Durability was also not considered.

The assessment is a worth source of information for architects and decision makers.

Yours sincerely,

Rozália Vaňová

Round 2

Reviewer 1 Report

Dear authors,
I really appreciate your improvements that have taken the article to the next level.
I must point out that Table 4 looks very strange - isn't that the last two column error?

I have to ask: in the article, you show the positive benefits of straw due to the absorption of heavy metals - but where do you get the certainty that heavy metals are not excreted from the wall? There are studies on this topic. Where do you get your assumptions from?
Here I would pay close attention to similar considerations.

Regards,

Author Response

Dear reviewer,

thank you for your report. 

I have checked the Table 4 and found no errors. The two last columns in the table refer to weight contribution within the structure. That means, for example the weight of reinforcing steel (construction material no. 1) in the construction is 241.49 kg, which is equal to 0.31 % of overall weight of construction.

The second issue on benefits of straw due to the absorption of heavy metals is as follows: 

We do not claim that heavy metals are excreted from the wall. That was not contained in the study. What you mean belongs to the B1 life cycle stage according to Table 1. The study only deals with the product stage from A1 to A3, so it assesses the environmental impact of construction material from raw material extraction until the manufacture of the material.

You can find evidence in the attached file. Negative signs represent the absorption.

Yours sincerely,

Rozália Vaňová

Reviewer 2 Report

The revised article “materials-1176878_v2” entitled: “Environmental Impact Assessment of Load Bearing Straw Bale Residential Building” has slightly been improved. There are still many shortcomings concerning the value of contribution, the presentation style and the topic of this paper, which have not been remediated. Perhaps is rather difficult to overcome the weaknesses of the paper, and therefore, the suggestion, unfortunately, is still to reject the paper.

Author Response

Dear reviewer,

thank you for your report. 

The topic of the study was changed to “Life Cycle Impact Assessment of Load Bearing Straw Bale Residential Building”, which we consider eloquent. We believe we made the best the fit the article for the special issue "Environmentally Friendly Materials in Construction". Changes are highlighted in the article.

Yours sincerely,

Rozália Vaňová